# Novel Thermostable Heparinase Based on the Genome of Bacteroides Isolated from Human Gut Microbiota

**DOI:** 10.3390/foods11101462

**Published:** 2022-05-18

**Authors:** Chuan Zhang, Leilei Yu, Qixiao Zhai, Ruohan Zhao, Chen Wang, Jianxin Zhao, Hao Zhang, Wei Chen, Fengwei Tian

**Affiliations:** 1State Key Laboratory of Food Science and Technology, Jiangnan University, Wuxi 214122, China; 13132551195@163.com (C.Z.); zhaiqixiao@sina.com (Q.Z.); 6200113144@stu.jiangnan.edu.cn (R.Z.); 7180112085@stu.jiangnan.edu.cn (C.W.); zhaojianxin@jiangnan.edu.cn (J.Z.); zhanghao61@jiangnan.edu.cn (H.Z.); chenwei66@jiangnan.edu.cn (W.C.); fwtian@jiangnan.edu.cn (F.T.); 2School of Food Science and Technology, Jiangnan University, Wuxi 214122, China; 3National Engineering Research Center for Functional Food, Jiangnan University, Wuxi 214122, China; 4Wuxi Translational Medicine Research Center and Jiangsu Translational Medicine, Research Institute Wuxi Branch, Wuxi 214122, China

**Keywords:** glycosaminoglycan, gut microbiota, *Bacteroides*, heparinase, enzymatic properties, thermostability

## Abstract

Among the nutrients available to the human gut microbiota, the complex carbohydrates and glycosaminoglycans are important sources of carbon for some of the species of human gut microbiota. Glycosaminoglycan (heparin) from the host is a highly preferred carbohydrate for *Bacteroides*. To explore how gut microbiota can effectively use heparin as a carbon source for growth, we conducted a screening of the Carbohydrate-Active enzymes (CAZymes) database for lytic enzymes of the PL13 family and Research Center of Food Biotechnology at School of Food Science and Technology of Jiangnan University database of *Bacteroides* to identify novel glycosaminoglycan-degrading bacterial strains. Four *Bacteroides* species (*Bacteroides eggerthii*, *Bacteroides clarus*, *Bacteroides nordii*, and *Bacteroides finegoldii*) that degraded heparin were selected for further studies. Analysis of the polysaccharide utilization sites of the four strains revealed that all of them harbored enzyme encoding genes of the PL13 family. Functional analysis revealed the activity of CAZymes in a medium containing heparin as the sole carbon source, suggesting their potential to degrade heparin and support growth. The four enzymes were heterologous expressed, and their enzymatic properties, kinetics, and thermal stability were determined. The lytic enzyme of *B. nordii* had high enzymatic activity and thermal stability. The features that cause this high thermal stability were elucidated based on an examination of the three-dimensional structure of the protein. Our findings provide an important theoretical basis for the application of glycosaminoglycans and glycosaminoglycan-degrading enzymes in the medical and biotechnology industries, and an important scientific basis for precision nutrition and medical intervention studies using gut microbiota or enzymes as targets.

## 1. Introduction

The human body is inhabited by numerous microorganisms, mainly residing in the mouth, skin, and gut. Of these organs, the human gut harbors a rich and diverse microbial community, accounting for approximately 80% of the total microorganisms that inhabit humans [1]. The human gut microbiota (HGM) is a large microbial community that is vital to human health [2,3,4]. The maintenance of HGM depends on its ability to use diet- and host-derived polysaccharides as a nutrient source [5,6]. Complex polysaccharides serve as the main carbon source for the HGM, and carbohydrate-active enzymes (CAZymes), which control the assembly and breakdown of these polysaccharides, are involved in the metabolism and utilization of polysaccharides by the gut microbiota, ultimately assisting the human digestive system with carbohydrates degradations [7,8]. The human gut microbiota is a remarkable source of enzymes, and new CAZymes are increasingly being discovered [9,10].

Glycosaminoglycans are the most stable source of nutrients for the gut microbiota and are the preferred high-utilization carbon source for *Bacteroides* [11,12]. The major glycan degraders in the HGM belong to the phylum *Bacteroidetes* [5,13,14]. Host heparin is a high-priority carbohydrate of *Bacteroides* in the HGM. Heparin is a polyanionic and heterogeneous polysaccharide of the glycosaminoglycan family that is widely present on the cell surface, extracellular matrix, and intracellular environment [15,16,17]. Heparin has various physiological functions, including participation in various physiological and pathological processes and the regulation of gut homeostasis [18,19,20]. Identifying the enzymes that can degrade heparin is of great significance for evaluating its role in the regulation of gut homeostasis. Bacterial heparinases (HepI) are polysaccharide lyases that specifically catalyze the β-eliminative reaction of glycosidic bonds between GlcNAc and GlcA/IdoA residues in heparin chains to produce oligosaccharides containing a 4,5-unsaturated uronic acid residue at the nonreducing end. Heparinases are indispensable tools for the structural and functional studies of heparin [21,22]. In recent years, the isolation of heparinase-producing intestinal bacteria has attracted increasing attention. To date, different *Bacteroides* species, including *B. thetaiotaomicron* strain WAL2926 [23] and *B. cellulosilyticus* [24], have been shown to produce heparinase. Furthermore, some enzymes have been identified and characterized. However, similar enzymes have not been isolated from *B. clarus*, *B. finegoldii*, *B. eggerthii*, and *B. nordii*.

In this study, we analyzed and annotated the data from the CAZymes database based on the whole genome sequence of the gut microbiota. Four uncharacterized heparin-degrading strains of *Bacteroides* were discovered. Further analysis revealed that all the four *Bacteroides* species could efficiently utilize heparin in vitro, and the four new heparinases had good enzymatic activity. Among them, heparinase from *B. nordii* was identified to have good thermal stability and high industrial application value. This study provides the base line for the extensive mining of gut microbiota encoding enzyme systems and the precise understanding of the molecular transformation mechanism of these enzymes, as well as for the establishment of a precise nutrition/medical system based on gut enzymes.

## 2. Materials and Methods

### 2.1. Strains, Plasmid, Culture Conditions, and Chemicals

All four strains (*B. clarus* FFJLY22K22, *B. finegoldii* FNMHLBE3K7, *B. eggerthii* FSDTAHCKB9, and *B. nordii* FTJS11K9) used in this study were screened and preserved at our research center. *Bacteroides* were cultured in brain heart infusion (BHI) broth at 37 °C in an anaerobic workstation [25]. pE-SUMO with a 6× His tag was employed as the overexpression vector. The biochemical reagents were purchased from Shanghai Biotech (Shanghai, China), and nuclease was purchased from Thermo Fisher.

### 2.2. Analysis of the Heparin-Degrading Enzyme Strains

Heparin lyase (EC 4.2.2.7) belongs to the polysaccharide lyase (PL) 13 family. The polysaccharide lytic enzyme 13 family was analyzed, with a total of 11 *Bacteroides* species (http://www.cazy.org/PL13.html) (5 November 2021). Seventeen species of *Bacteroides* were screened in a previous study by our group. In this study, we found that there were 10 species of *Bacteroides* that have never been reported to utilize heparin in the CAZymes database, when compared with Bacteroides harboring PL13 family heparinases (Appendix A). Genome-wide analysis was performed using the 10 *Bacteroides* strains, particularly on the polysaccharide utilization site (*PULs*) against heparin, to further identify novel strains that can utilize heparin.

### 2.3. Genome Sequencing, Data Assembly, and CAZymes Annotation

The sketch genome of *Bacteroides* was sequenced using the Illumina Hiseq × 10 platform. Thereafter, a 2 × 150 bp paired library and a paired library with an average read length of approximately 400 bp were constructed. The reads were assembled using SOAPdenovo and the local internal gaps were filled using GapCloser [26].

CAZymes in the Prokka-annotated genome of *Bacteroides* were annotated using the dbCAN2 metaserver (http://bcb.unl.edu/dbCAN2/ (accessed on 4 February 2022)) [27,28], while carbohydrate utilization genes were annotated using the Carbohydrate-Active Enzyme Database [29]. The annotation results were clustered using HemI [30].

### 2.4. Sequence Alignment and Evolutionary Tree Construction

An initial sequence alignment was performed using ClustalW [31] Thereafter, the result was, displayed as a figure prepared using ESPript 3.0 [32]. The phylogenetic tree of the PL13 gene family and other heparinases was constructed using MEGA6.0 software and the neighbor-joining method, with a bootstrap value of 1000.

### 2.5. Utilization of Heparin by Bacteroides and Growth Curves

*Bacteroides* strains were inoculated in basal medium (BHI 38.5 g/L, hemin 0.02 g/L, and Vitamin K1 0.0025 g/L) under anaerobic conditions at 37 °C for 18–24 h and washed three times with sterile PBS. The inoculum was adjusted to 600 nm = 0.6 and diluted at a ratio of 1:10 in a single carbon source medium (L-cysteine 0.5 g/L, hemin 0.02 g/L, NH_4_Cl 1 g/L, CaCl_2_·2H_2_O 0.015 g/L, Na_2_HPO_4_ 6 g/L, KH_2_PO_4_ 3 g/L, NaCl 0.5 g/L, MgSO_4_·7H_2_O 0.25 g/L, FeSO_4_·7H_2_O 0.004 g/L, V_k1_ 0.0025 g/L, V_B12_ 0.005 g/L, and carbon source 5 g/L) [33]. Of note, heparin was the only carbon source. After 72 h of anaerobic incubation at 37 °C, centrifugation was performed at 10,000× *g* for 15 min at 4 °C to obtain the supernatant.

The growth curve was determined using the multifunctional detection enzyme standard, Infinite F50 (Tecan Life Sciences, Mannedorf, Switzerland), to derive the OD_600_ values [34] automatically after every 1 h and continuously for 72 h. The growth curves of different strains of *Bacteroides* spp. on medium consisting of heparin as the sole carbon source were evaluated.

### 2.6. Determination of Heparin Consumption

The total carbohydrate concentration was determined using the phenol-sulfuric acid method [35,36]. Three parallel trials were conducted with carbohydrate content as a control. The results are expressed as the mean value of carbohydrate residues at 0 h of fermentation in the control group.

### 2.7. Cloning, Expression, and Purification of Heparinase

The coding gene sequence of heparinase I *B. clarus* (Bc-HepI), *B. finegoldii* (Bf-HepI), *B. eggerthii* (Be-HepI), and *B. nordii* (Bn-HepI) were optimized according to the codon usage table of *E. coli* and synthesized by Genewiz (Suzhou, China). *B. thetaiotaomicron* (Bt-HepI) was employed as the positive control in this study. The synthesized gene sequences were subjected to PCR using the primers listed in Table 1, and the resulting products were inserted into the *BamH* I and *Xho* I sites of the pE-SUMO vector via homologous recombination. PCR reaction conditions were: 94 °C for 2 min, 94 °C for 30 s, 60 °C for 30 s, 72 °C for 30 s, 32 cycles, 72 °C for 10 min and 4 °C for 10 min. PCR products were subjected to cut-gel recovery (GeneJET Gel Extraction Kit, thermo scientific). The pE-SUMO vector was double digested using *BamH* I and *Xho* I and cut-gel recovery was performed. The target gene and vector were then ligated using a homologous recombination kit (ClonExpress II One Step Cloning Kit). Then, 10 uL of ligation product was transformed into 100 uL of receptor cells.

Thereafter, plasmids were sequenced (GENEWIZ, Suzhou) and transformed into the expressed strains (*E. coli* Rosetta (DE3)). The recombinant strain carried the pE-SUMO-HepI plasmid and was cultured for 12–15 h at 37 °C on LB agar plates containing 34 μg/mL of chloramphenicol and 50 μg/mL of kanamycin. The positive monoclonal was picked and inoculated in 5 mL of liquid LB medium containing the corresponding resistance and incubated overnight at 37 °C, 220 rpm. The seed culture (1%) was then inoculated into 100 mL of LB medium with antibiotics and cultivated at 37 °C and 220 rpm until the optical density at 600 nm (OD_600_) reached 0.6. The cells were induced with 0.6 mM isopropyl-β-D-thiogalactopyranoside (IPTG) at 30 °C for 9 h. Thereafter, the cells were harvested by centrifugation at 8000 rpm for 10 min at 4 °C, washed twice with Tris–HCl buffer (20 mM Tris, 200 mM NaCl, 5 mM CaCl_2_, pH 7.4) and resuspended in 30 mL of the same buffer. Sonication (sonication 3 s, pause 4 s, 300 W) was performed on ice for 10 min to disrupt the cells. Cell debris was separated via centrifugation at 12,000 rpm for 20 min at 4 °C. The expressed HepI was the soluble protein in the supernatant and was resolved by using 12% SDS-PAGE [37]. The target proteins stained with Coomassie Brilliant Blue R-250 were visualized using an Odyssey Infrared Imaging System (Gene Company, Beijing, China). Protein purification steps: Firstly, the supernatant containing the target protein was filtered using a 0.45 mm filter membrane and then loaded onto a Hi-Trap metal-chelating affinity column. Secondly, it was washed with 10 mL of washing buffer (50 mM Tris, 300 mM NaCl, 5 mM CaCl_2_, 50 mM imidazole, pH 7.4) to remove non-target proteins. Finally, 3–5 mL elution buffer (50 mM Tris, 300 mM NaCl, 10 mM CaCl_2_, 150 mM imidazole, pH 7.4) was applied to obtain the purified target protein. The expression yield of HepI was analyzed by SDS-PAGE, and measured using Bradford’s method [38].

### 2.8. Effects of Temperature and pH on Enzyme Activity

Enzyme activity was measured at 20, 25, 30, 35, 40, 45, 50, 55, and 60 °C to determine the optimum reaction temperature. Enzyme activity at the optimum temperature was defined as 100%, and the relative enzyme activity of HepI I was determined. The optimal pH of the pure enzyme was measured by determining the enzyme activity at pH values ranging from 3 to 10. Enzyme activity at optimum pH was defined as 100%, and the relative enzyme activity of HepI was determined. The optimal concentrations of heparin for the pure enzyme was measured by determining the enzyme activity at various concentrations of heparin (0, 2, 3, 5, 10, 15, 20, and 25 mg/mL). Enzyme activity at the heparin concentration of 25 mg/mL was defined as 100%, and relative enzyme activity of HepI was determined.

### 2.9. Enzyme Activity Determination and Kinetic Parameters

HepI activity was measured using the 232 nm method [39]. The reaction was carried out with heparin as the substrate at 30 °C. Heparin degradation was detected at 232 nm using a UV-3200 spectrophotometer with a molar extinction coefficient of 3800 M^−1^ cm^−1^.

The kinetic parameters of Bc-HepI, Bf-HepI, Be-HepI, and Bn-HepI were determined at optimum temperature and pH. The *Km* and *Vmax* values were calculated after fitting.

### 2.10. Thermostability of HepI

#### 2.10.1. Determination of Half-Life

The half-lives (*t*_1/2_) of purified Bc-HepI, Bf-HepI, Be-HepI, and Bn-HepI were determined via incubation at 40 °C and 50 °C for different durations. The half-life (*t*_1/2_) was calculated according to the first-order deactivation function: ln (residual activity) = −*k*_D_t; *t*_1/2_ = *ln*2/*k*_D_. *k*_D_ is the deactivation rate constant.

#### 2.10.2. Determination of $T_{50}^{15}$

$T_{50}^{15}$, the temperature at which the enzymes lose 50% activity after incubation for 15 min, was determined by adding of 50 μL of purified enzyme (1 mg mL^−1^) into a row of a 96-well plate. The 96-well plate was incubated at temperatures ranging from 20 to 50 °C for 15 min, frozen, and assayed for the residual activity of Bc-HepI, Bf-HepI, Be-HepI, and Bn-HepI.

#### 2.10.3. Determination of Melting Temperature (*Tm*)

*Tm* was measured using a capillary differential scanning calorimeter (MicroCal VP-Capillary DSC, GE Company, Fairfield, CT, USA). Samples and buffers were degassed before testing. GE MicroCal thermo vac was used to degas the samples and buffers at 10 °C for 5 min. The sample concentration was 0.4 mg/mL, and the buffer concentration was 50 mmol/L Tris-HCl (pH 7.4). The sample and buffer were added to the corresponding sample and buffer pool, respectively, with a sample volume of 400 μL. The measuring temperature ranged from 20 to 90 °C, and the heating rate was 1 °C/min. MicroCal VP-Capillary DSC software 2.0 (Westborough, MA, USA) was used for control and sample data acquisition, and Origin^TM^ software (Northampton, MA, USA) was used for data processing. In the measurement results, the ordinate is Cp, the abscissa is the temperature, the abscissa corresponding to the highest point of the unfolding curve is the protein unfolding temperature *Tm*, and the integral value of the unfolding curve and the corresponding temperature is the enthalpy value Δ*H.* The Tehoff enthalpy value Δ*Hv* was determined using the peak shape of the unfolding curve.

#### 2.10.4. Determination of the Secondary Structure Using Circular Dichroism Spectroscopy

Dialysis of the purified enzymes (Bc-HepI, Bf-HepI, Be-HepI, and Bn-HepI) was performed to remove as much salt ion concentration as possible. The sample concentration was 0.5 mg/mL. Briefly the sample was added to a quartz cuvette and placed on a circular dichroic spectrometer for detection. The specific parameters were as follows: the optical path of the sample cell was 1 mm, the step length was 0.1 nm, the scanning range was 190–250 nm, and the scanning speed was 120 nm/min. The Origin Pro 8.5 software (Northampton, MA, USA) was used to construct a map and the CD Pro software was used to fit the composition and content of the secondary structure in the protein.

### 2.11. Homology Modeling and Molecular Docking

Based on the known crystal structure of *B. thetaiotaomicron* heparinase (PDB code:3IKW), SWISS-MODEL was used to construct a 3D model of heparinase [23]. The stereochemistry of the structure was determined using UCLA’s Structural Analysis and Verification Server. Molecular docking of the heparinases and heparin was performed using AutoDock. Finally, all structure maps were generated using PyMOL (www.pymol.org) (accessed on 4 February 2022) [40].

### 2.12. Statistical Analysis

The experiments were conducted three times in parallel, and the results are expressed as mean ± standard deviation. A *t*-test was used for the statistical analysis.

## 3. Results

### 3.1. Analysis of Genomes, CAZymes, and PULs

By performing whole genome analysis, gene assembly, and functional gene annotation, only four of the ten strains screened in the previous stage were found to contain heparinase genes (*B. eggerthi*, *B. clarus*, *B. nordi*, and *B. finegoldi*). To determine the utilization of heparin by *Bacteroides,* their genome was annotated with CAZymes using the dbCAN2 meta-server pipeline. The CAZymes distribution of the common intestinal microorganisms was examined (Figure 1a), which revealed five carbohydrases, including glycoside hydrolase (GH), carbohydrate esterase (CE), polysaccharide lyase (PL), glycosyltransferase (GT), auxiliary activities (AA) and carbohydrate-binding module (CBM) (Figure 1b). A closer examination of the screen of CAZymes in the quad-screened genomes of the *Bacteroides* revealed that at least one polysaccharide lyase per strain was predicted to specifically target heparin, most of which contained a signal peptide.

The heparin *PULs* of the four strains were predicted and analyzed (Figure 1c). As a result, these four strains were predicted to potentially utilize heparin efficiently as a carbon source.

### 3.2. In Vitro Fermentation of Heparin Polysaccharide

Based on the results of the above genomic and CAZymes data analyses, it was determined that the four *Bacteroides* strains may utilize heparin. The strains were cultured in a medium with heparin as the only carbon source and fermented for 72 h. Growth curves were measured and significant changes in heparin content were observed. As shown in Figure 2a, the growth curves of the four strains of *Bacteroides* based on heparin were generated, and the OD_600_ values were measured at 1 h intervals. The four strains could grow on a medium consisting of heparin as the sole carbon source. Among these, *B. nordii* displayed the best growth conditions. As shown in Figure 2b, *B. nordii* could best utilize heparin, and the remaining three strains were slightly less capable of utilizing heparin. Significant differences in heparin consumption by *B. nordii* were observed relative to heparin consumption by *B. clarus* and *B. finegoldii*. The results of the polysaccharide consumption measurements were consistent with those of the growth curve. Therefore, all four strains of *Bacteroides* obtained using genomic and CAZymes analyses could effectively utilize heparin.

### 3.3. Multiple Sequence Alignment and Evolutionary Tree

In this study, we mined the genes corresponding to heparinase in four species of *Bacteroides* spp. through a pretest genome bioanalysis and a pre-analysis of genomic raw letters. The amino acid sequences of the heparinase genes of *B. thetaiotaomicron*, *B. clarus*, *B**. eggerthii*, *B. nordii*, and *B. finegoldii* were compared, and the results are shown in Figure 3a. Comparative analysis of the amino acid sequences of heparinase from *B. thetaiotaomicron* revealed that the amino acid sequence similarity of these four heparinases was 80% and the four screened strains of *Bacteroides* contained the heparinase gene. Evolutionary tree analysis revealed that the heparinase genes of the four strains of *Bacteroides* slightly differed in terms of proximity (Figure 3c). The three-dimensional (3D) structures of the four heparinases were simulated using homology modeling. A 3D conserved sequence analysis structure map was constructed by performing multiple sequence alignments (Figure 3b). Based on the map, the active centers of the proteins were highly conserved; however, large differences were found in the protein surfaces. The active centers of the proteins are closely related to the catalytic activity of the enzymes, whereas the surface amino acid residues of the enzymes are closely related to the thermal stability of the enzymes. Based on the above results, a new source of heat-stable heparinase can be obtained. Therefore, heterologous fusion expression of the four screened heparinase genes was performed.

### 3.4. Fusion Expression and Purification of Heparinase

The recombinant expression strains, Bc-HepI, Bf-HepI, Be-HepI, and Bn-HepI, were successfully constructed. The results of the construction of the fusion expression plasmids are shown in Appendix A. The fusion of SUMO-Tag at the N-terminal end was used to further improve the water solubility of the target protein and promote its correct folding. The purity of heparinase was analyzed using 12% SDS-PAGE (Appendix A; only the purification results of Bn-HepI are shown). Based on gray value analysis, the protein concentrations of Bc-HepI, Be-HepI, Bf-HepI, and Bn-HepI were 0.667, 0.550, 0.705, and 0.667 mg/mL, respectively (Table 2). The specific enzyme activity of Bn-HepI was higher than that of the positive control (Bt-HepI). The physical properties of the four enzymes are listed in Table 3.

### 3.5. Enzymatic Characterization

The enzymatic properties of the four heparinases were determined to evaluate the biological activities of the enzymes. The optimal temperature and pH of heparinase activity from different sources did not consistently show differences (Figure 4). The optimal reaction temperature and pH of Bc-HepI, Be-HepI, Bf-HepI, and Bn-HepI, and the maximum enzyme activity of each mutant at different temperatures and pH values were set to 100%. The relative activities of each enzyme at different temperatures and pH levels were calculated to determine the optimal reaction temperature and pH for each enzyme (Figure 4). The optimal reaction temperatures for Be-HepI, Bc-HepI, and Bn-HepI were 30, 35, and 40 °C, respectively. However, the optimum reaction pH of Bc-HepI, Be-HepI, and Bf-HepI remained constant at 7, while that of Bn-HepI was 8. The amino acid composition of the four enzymes was analyzed and Bn-HepI was found to have more basic amino acids than the other three enzymes, which may be the reason for its higher optimal pH relative to that of the other enzymes. As analyzed previously, these four enzymes have distinct differences in surface amino acid residues, thereby exhibiting different enzymatic characterizations.

### 3.6. Thermal Stability and Enzyme Kinetics

To verify the thermal stability of the enzymes, the half-lives of each enzyme at 40 °C (Optimum temperature) and 50 °C (Deactivation temperature) were determined. Compared with Be-HepI, the half-lives of Bc-HepI, Bf-HepI, and Bn-HepI at 40 °C were, respectively, 71.29%, 82.18%, and 141.58% higher than that of Be-HepI. The half-life of Be-HepI was 101 min (Figure 5a). The half-lives of Bc-HepI, Bf-HepI, and Bn-HepI at 50 °C were 32.65%, 61.22%, and 116.33% higher than that of Be-HepI, respectively (Figure 5b).

The temperature at which enzyme activity was reduced by half after 15 min of incubation at different temperatures was determined. The higher the corresponding temperature, the better was the thermal stability. The temperature corresponding to the different enzymes was markedly varied (Figure 5). Be-HepI corresponds to the lowest temperature of 25.8 °C. Compared with Be-HepI, the temperatures of Bc-HepI, Bf-HepI, and Bn-HepI were higher by 8.3, 12, and 18 °C, respectively (Table 4). The melting temperatures of the enzymes, which serve as a reliable indicator of protein stability, were determined. *Tm* can accurately reflect the thermal stability of proteins and be used to screen out enzymes with good thermal stability, quickly and efficiently. As shown in Appendix A, Be-HepI had the lowest *Tm* value of 38.97 °C. Compared with Be-HepI, the *Tm* values of Bc-HepI, Bf-HepI, and Bn-HepI were higher by 0.45, 2.82, and 5.82 °C, respectively. The activity and thermal stability of Bn-HepI were better than those of the positive control (Table 4).

Proteins have both flexible and rigid structures, with the flexible structure providing enzyme activity and the rigid structure conferring enzyme stability. Accordingly, we measured the thermal stability and kinetics of the enzyme to determine the enzyme activity. The *kcat*/*Km* scores of Be-HepI, Bc-HepI, Bf-HepI, and Bn-HepI were 279.3, 369.5, 404.3, and 799.6, respectively. Bc-HepI, Bf-HepI, and Bn-HepI had 32.30%, 44.75%, and 186.29% higher *kcat*/*Km* than Bc-HepI, respectively (Figure 6). Based on the above results, Bn-HepI not only had a high catalytic efficiency, but also a high thermal stability. Such a finding indicates that the gut microbial genome-based mining of novel enzyme resources used in this study is feasible and provides an important direction for the further development of gut microbial resources. However, to study the thermal stability of enzymes, the 3D structure of proteins must be further investigated to analyze the underlying causes.

### 3.7. Structural Analysis of Heparinase

To explore the differences in thermal stability between different enzymes, the spatial 3D structures of Bc-HepI, Be-HepI, Bf-HepI, and Bn-HepI were systematically analyzed. Structural simulations were performed for the four heparinases, and structural superimposition maps with substrates were produced to further analyze the structural differences between the different enzymes (Appendix A). The core activity pockets of the four enzymes were extremely similar, with some differences in the surface structure of the proteins. Structural stereochemistry was analyzed using the UCLA Structural Analysis and Validation server (http://services.mbi.ucla.edu/SAVES/ (accessed on 4 February 2022)) (Appendix Ab). All structures were found to meet the stipulated requirements. Structural analysis revealed the existence of cation-π stacking interactions (bond energy distance of 2.0 Å) between the amino acids Tyr230 and Glu254 in the peripheral flexible region of the Bn-HepI protein. The cation-π stacking interactions between Tyr230 and Glu254 can effectively stabilize this flexible region and thus improve the thermal stability of Bn-HepI (Figure 7). In contrast, Bc-HepI, Be-HepI, and Bf-HepI do not have corresponding cation-π stacking interactions in this region because their spatially corresponding amino acid is tryptophan (Trp), instead of tyrosine (Tyr). As the spatial stretching directions of the two amino acids are different, the local spatial conformation also differs, which may serve as the main reason for the good thermal stability of Bn-HepI.

## 4. Discussion

The microorganisms that reside in humans are mainly bacteria, with the thick-walled bacteria of the Bacteroidetes accounting for the highest proportion [41,42,43]. Although these bacteria play an important role in human physiology, how they compete effectively in this intense ecosystem is poorly understood [44]. Despite providing a stable source of carbon, the human gut contains few phyla, suggesting the selectivity of bacteria regarding their hosts [41]. Previous studies have shown that glycosaminoglycans play an important role in intestinal homeostasis by acting as a stable source of nutritional carbon for host intestinal microorganisms and can be used as a preferential carbon source for the genus, *Bacteroides* [45,46].

Heparin is a glycosaminoglycan with a sulfated and repeating-unit structure [47,48], and intestinal microorganisms can use glycosaminoglycans [36,49,50]. Many CAZyme genes are present in the gut microbiome, with different levels of gene activities in different species [51]. The heparin-degrading bacterium *B. stercoris* HJ-15 has been isolated from human feces [52]. Some studies have shown that heparin can modulate intestinal microorganisms and increase the adhesion of *Lactobacillus* spp. [53]. However, to date, the degradation of heparin in the intestine has not been systematically explained, and the functional enzyme genes involved in heparin degradation by intestinal microorganisms have not been well elucidated. The lack of such critical information has markedly hindered further studies on heparin. Therefore, strains that can degrade heparin, based on intestinal microorganisms, must be studied and novel heparinases must be explored.

In this study, we mined four species (*B. clarus*, *B. eggerthii*, *B. finegoldii*, and *B. nordii*) (Table 1) with unreported heparin degradation ability by comparing and analyzing the CAZymes database and the strain library established at our center. Further research, whole-genome sequencing, and functional annotation of common microorganisms (*Akkermansia muciniphila*, *Clostridium butyricum*, *Bifidobacterium*, and *Lactobacillus*) in the human gut were performed, and their CAZymes were further analyzed (Figure 1). *Bacteroides* was identified as the most abundant source of the carbohydrate glycosidase gene, followed by *Clostridium butyricum*, and *Akkermansia muciniphila*. The number of CAZymes was different in the different species of *Bacteroides*. Therefore, individual microbial differences may lead to individual differences in heparin degradation ability. To investigate the mechanism of this phenomenon, growth curve measurements and heparin consumption measurements were carried out using heparin as the sole source of carbon. The results verified the previous conjecture that the heparin-degrading strain mined from the genomic analysis could grow with heparin as a carbon source. Growth curve and polysaccharide consumption measurements revealed that of the four species studied, *B. nordii* was the optimal species (Figure 2).

Heparinase belongs to the PL13 family of polysaccharide lytic enzymes. The known strains of bacteria hosting the PL13 family of genes were screened with the four strains of *Bacteroides* to construct an evolutionary tree. These four *Bacteroides* were found to be distant from the remaining three strains in terms of their affinity for *B. nordii* (Figure 3a). The amino acid sequences of the four heparinases were analyzed by multiple sequence alignment for structural conservativeness. The heparinases of *B. nordii* were found to slightly differ in the amino acid residues on the protein surface, However, the active pocket regions of the four heparinases remained almost identical. These findings suggest that Bn-HepI may have a large variation in thermal stability (Figure 3b).

To explore the activity and thermal stability of the four novel heparinases of *Bacteroides* origin, heterologous expression of the relevant genes was performed (Appendix A). SDS-PAGE revealed that all four heparinases were successfully solubilized and expressed (Appendix A). The enzymatic properties and enzyme kinetics of the four enzymes were determined, and Bn-HepI was identified to have the highest catalytic efficiency. To date, only few reports on the thermal stability of HepI have been published. Some studies have reported the inactivation mechanism of HepI, such as the thermal inactivation mechanism of the maltose-binding protein (MBP)-HepI fusion protein, in which the MBP tag improved the thermal stability of the enzyme via dithiotreitol [54]. The various protectants were found to be small molecular compounds, which will cause difficulties in the separation and extraction of final products. To date, the thermal stability of heparinase and the determination of its half-life at different temperatures have not been reported in a full article. To explore the thermal stability of the four heparinases, detailed measurements, including the half-life at 40 °C and 50 °C, half-inactivation temperature and the melting temperature of the enzymes, were performed in this study. Based on the results, Bn-HepI has a higher thermal stability than Bc-HepI, Be-HepI, and Bf-HepI, with a half-life of 10.6 min at 50 °C. The melting temperature of Bn-HepI was also significantly higher than that of the other proteins.

There are few reports on the structure of heparinases. At present, only the structure of HepI, including the active region of the enzyme and the binding region of Ca^2+^ ions in *Bacteroides* have been analyzed [23]. However, the thermal stability of HepI has not been evaluated. In this study, the thermal stability of heparinase was further assessed by simulating the protein structure and docking with the substrate (Appendix A). π-stacking is a weak interaction caused by the conjugation of the π-electron system in a special spatial configuration. At present, the widely accepted forms of π-stacking are π–π, cation–π, anion–π, halogen bond–π, and CH–π stacking. π-stacking is widely found in biological macromolecular systems and often plays a key role in the maintenance of specific structures of proteins and protein thermal stability [55,56]. Structural analysis of Bn-HepI revealed a cation–π conjugation between the amino acid residues, Tyr230 and Glu254. The other three proteins did not interact with each other.

## 5. Conclusions

In summary, a novel glycosaminoglycan lyase, Bn-HepI, with good enzymatic activity and thermal stability was found in a strain of *B. nordii* from the gut microbiota of healthy humans via gut microbial genome analysis and fusion expression. This study provides an effective method for mining functional enzyme genes from the gut microbiota and for exploring them further for industrial applications. Understanding the relevant heparin-degrading strains and enzyme genes lays the scientific foundation for the elucidation of the role of glycosaminoglycans in gut homeostasis and serves as a promising reference for further research and the development of next-generation probiotics.

## Figures and Tables

**Figure 1 foods-11-01462-f001:**
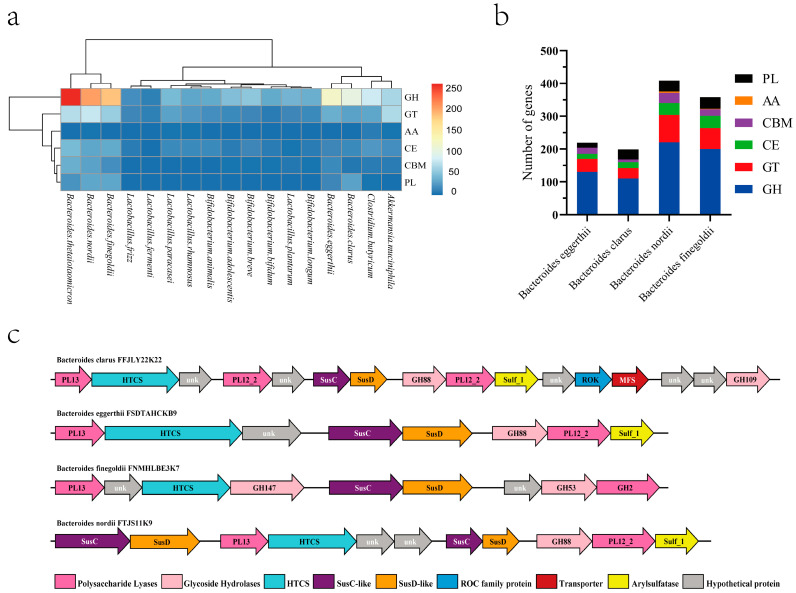
Genome-wide carbohydrase analysis of the gut microbiota. (**a**) Cluster heatmap of CAZymes of common intestinal microorganisms. (**b**) Specific statistics of the CAZymes of *B. eggerthii*, *B. clarus*, *B. nordii*, and *B. finegoldii*. (**c**) *PULs* of *B. eggerthii*, *B. clarus*, *B. nordii*, and *B. finegoldii*.

**Figure 2 foods-11-01462-f002:**
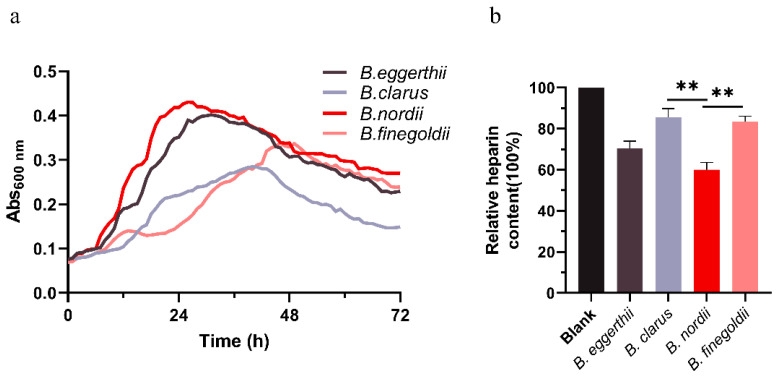
Growth curves and polysaccharide consumption of *B**. eggerthii*, *B. clarus*, *B. nordii*, and *B. finegoldii* based on heparin. (**a**) The growth of the strains was assessed by measuring the optical density at OD 600 nm over 72 h. Measurements were performed at 1 h intervals using the automatic detection function of the instrument. (**b**) Polysaccharide consumption was determined using the sulfuric acid-phenol method, and the polysaccharide content measured at 0 h of fermentation was set at 100%. **, *p* < 0.01.

**Figure 3 foods-11-01462-f003:**
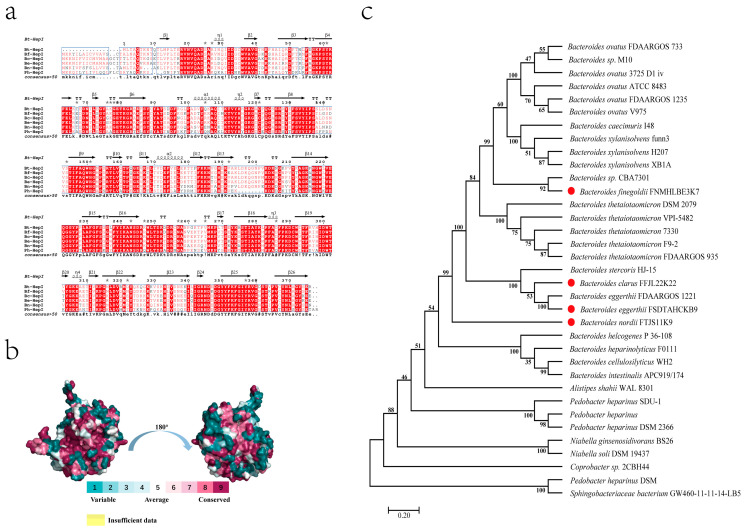
Multiple sequence alignment, conserved structural domains, and evolutionary tree analysis. (**a**) Structural sequence comparisons of Bf-HepI, Bc-HepI, Bc-HepI, and Bn-HepI forming secondary structure residues are highlighted in the Bt-HepI sequence. Identical and similar amino acid residues are indicated by white letters on a red background and black letters on a yellow background. The data were produced using ESPript. (**b**) Protein conserved structural domains are analyzed different numbers correspond to different colors, indicating different levels of conservativeness. Higher numbers indicate higher conservativeness. (**c**) An evolutionary tree of related enzyme genes was constructed using MEGA6.0, with red circles indicating the origin of the four enzymes in this study.

**Figure 4 foods-11-01462-f004:**
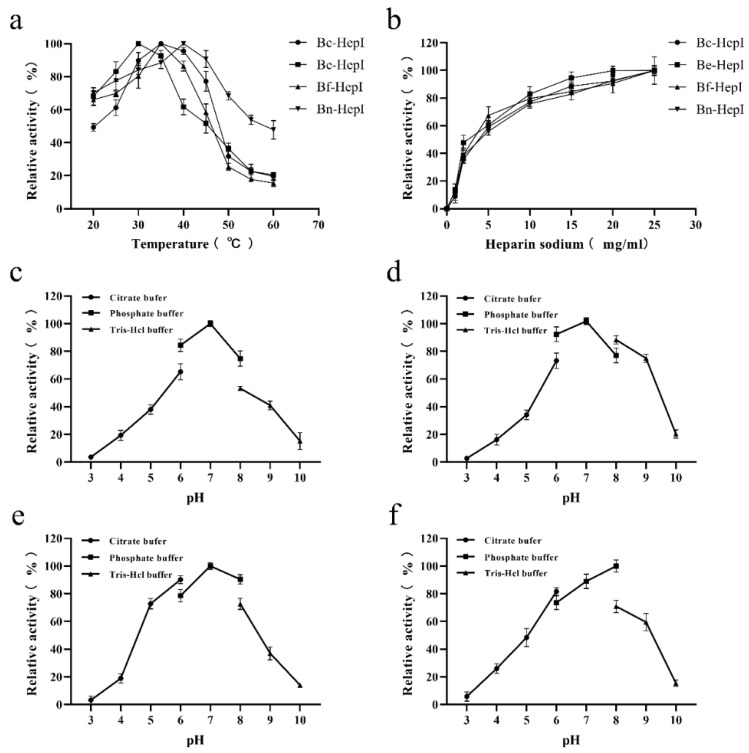
Effects of temperature (**a**), concentrations of heparin (**b**), and pH (**c**–**f**) on the activities of Bc-HepI (**c**), Be-HepI (**d**), Bf-HepI (**e**), and Bn-HepI (**f**).

**Figure 5 foods-11-01462-f005:**
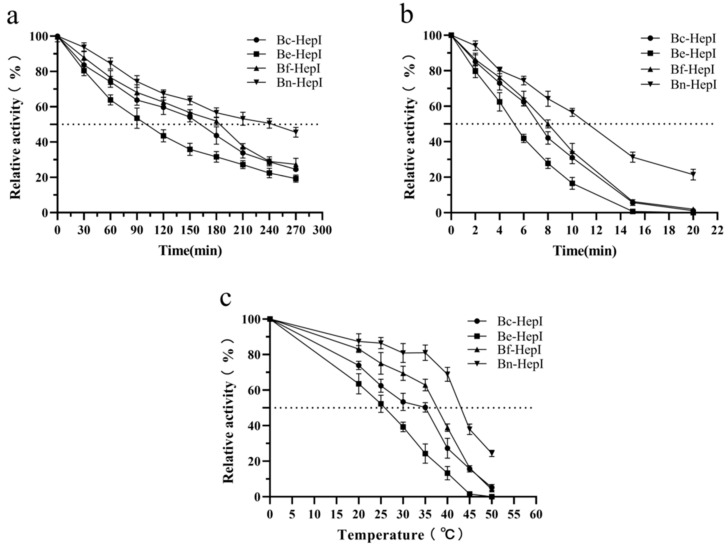
Half-life and $T_{50}^{15}$ analyses. (**a**,**b**) Temperatures of (**a**) 40 °C and (**b**) 50 °C were used in the analyses. The initial enzyme activity of different enzymes was set to 100%, and the enzyme activity of different treatment times was compared to obtain the relative enzyme activity. (**c**) The temperature at which the enzyme loses 50% of its activity after 15 min of incubation ($T_{50}^{15}$). Three independent experiments were performed in parallel for each set of data, and statistical analyses and graphing were performed.

**Figure 6 foods-11-01462-f006:**
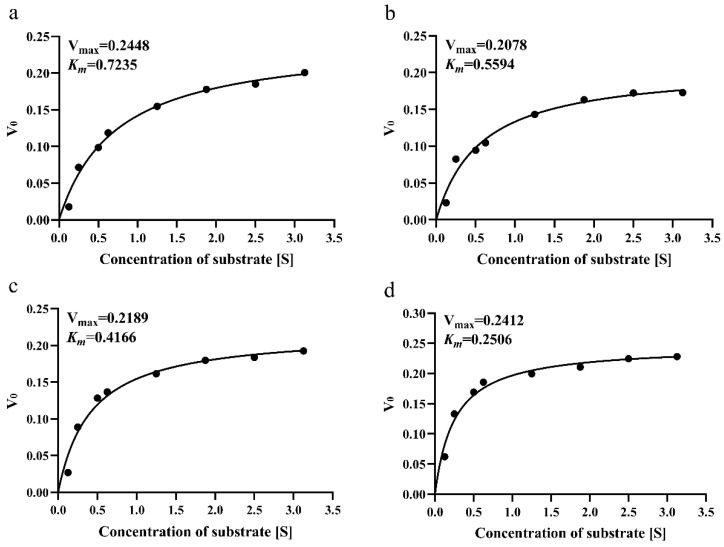
Enzymatic kinetics of Bc-HepI (**a**), Be-HepI (**b**), Bf-HepI (**c**), and Bn-HepI (**d**), including the *Km* and *Vmax*.

**Figure 7 foods-11-01462-f007:**
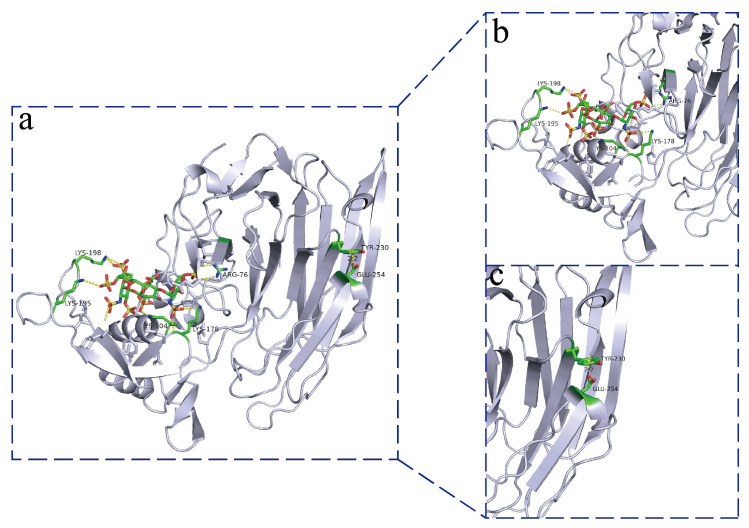
Structural analysis of Bn-HepI enzymes. (**a**) Molecular docking and thermal stability of the overall structure. (**b**) The interactions between ligands and proteins after molecular docking. (**c**) In Bn-HepI, a side-chain aromatic nucleus (Tyr230) appears above Glu254, which has a conjugated effect by forming cation-π stacking interactions between the protonated amino ions of Glu254. The co-action of cation-π stacking caused this flexible region to be more stable in Bn-HepI.

**Table 1 foods-11-01462-t001:** Oligonucleotide primer pairs used to construct the clone expression vector.

S. no.	Enzyme	Primer Sequences ^a^	Tm (°C)
1	Bc-HepI	F: 5′-*GAACAGATTGGAGGT*GGATCCACCGCGCAGACCAAAGGC-3′ R: 5′-*GTGGTGGTGGTGGTG*CTCGAG**TTA**TTCTTCTTTATAGCCCGCCAGG-3′	58–63
2	Be-HepI	F: 5′-*GAACAGATTGGAGGT*GGATCCACCGCGCAAGTGAAAAACGC-3′ R: 5′-*GTGGTGGTGGTGGTG*CTCGAG**TTA**TTCTTCTTTATAGCCCGCCAGG-3′	58–62
3	Bf-HepI	F: 5′-*GAACAGATTGGAGGT*GGATCCAACGCGCAGACCAAAAACACG-3′ R: 5′-*GTGGTGGTGGTGGTG*CTCGAG**TTA**TTTTTCGCTATAGCCCGCCAGG-3′	59–62
4	Bn-HepI	F: 5′-*GAACAGATTGGAGGT*GGATCCCAGAACGCGAAACTGATTCCGC-3′ R: 5′-*GTGGTGGTGGTGGTG*CTCGAG**TTA**GTTTTCGCTATAGCCCGCCAGG-3′	60–62

^a^ Restriction endonuclease sites are shown in underlined. The termination codon is shown in bold. Italics indicate the homologous sequence of the vector. Normal font indicates the specific primer for the target gene.

**Table 2 foods-11-01462-t002:** Enzyme activity and yield of Bc-HepI, Be-HepI, Bf-HepI, Bn-HepI and Bt-HepI.

Enzyme	TP * (mg/mL)	TEA * (IU/mL)	SA * (IU/mg)
Bc-HepI	0.667 ± 0.02	181.58 ± 3.96	272.23
Be-HepI	0.550 ± 0.04	178.84 ± 4.21	325.16
Bf-HepI	0.705 ± 0.01	214.21 ± 5.44	303.84
Bn-HepI	0.667 ± 0.03	240.42 ± 10.61	360.45
Bt-HepI	0.613 ± 0.02	192.44 ± 1.35	313.93

* TP, total protein; TEA, total enzyme activity; SA, specific activity.

**Table 3 foods-11-01462-t003:** Analysis of the physical properties of enzymes.

Enzyme	Number of Amino Acids (aa)	**Molecular Weight (KDa)**	**Theoretical pI**
Bc-HepI	374	42.4	8.57
Be-HepI	374	42.4	9.23
Bf-HepI	374	42.3	9.09
Bn-HepI	369	41.8	8.48
SUMO-Tag	108	12.4	5.71

**Table 4 foods-11-01462-t004:** Kinetic parameters and stability properties.

Enzyme	** *K* ** **m** **(mM)**	** *V* ** **max** **(mM/min)**	***k*****cat (**/s**)**	***k*****cat/*****K*****m** (/s/**mM**)	** *t* ** ** _1/2_ ** **(40 °C) (min)**	** *t* ** ** _1/2_ ** **(50 °C) (min)**	** *Tm* ** **(°C)**	$T_{50}^{15}$
Bc-HepI	0.72 ± 0.1	0.24 ± 0.04	201.1 ± 4.9	279.3 ± 3.8	173 ± 5.4	4.9 ± 0.5	39.42	34.1 ± 1.1
Be-HepI	0.56 ± 0.09	0.21 ± 0.06	206.9 ± 6.3	369.5 ± 6.3	101 ± 3.8	6.5 ± 0.4	38.97	25.8 ± 1.5
Bf-HepI	0.42 ± 0.1	0.22 ± 0.06	169.8 ± 5.1	404.3 ± 8.5	184 ± 4.7	7.9 ± 0.3	41.79	37.8 ± 1.8
Bn-HepI	0.25 ± 0.07	0.24 ± 0.05	199.9 ± 4.6	799.6 ± 0.1	244 ± 6.3	10.6 ± 0.4	44.79	43.8 ± 1.7
Bt-HepI	0.26 ± 0.02	0.20 ± 0.05	179.9 ± 4.6	692.0 ± 0.1	148 ± 1.2	6.6 ± 0.3	40.36	39.5 ± 1.2

## Data Availability

The whole genome of the strain were deposited in the National Center for Biotechnology Information (NCBI) under the dataset identifiers PRJNA786701, PRJNA786705, PRJNA786707 and PRJNA786710. Data can be accessed via https://www.ncbi.nlm.nih.gov/ (accessed on 4 February 2022) using the dataset above identifiers.

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
