# Peer review of "Novel Thermostable Heparinase Based on the Genome of Bacteroides Isolated from Human Gut Microbiota"

_foods, 2022, doi:10.3390/foods11101462_

Round 1
Reviewer 1 Report
Foods-1606377
Article
A novel thermostable glycosaminoglycan lyase based on the genome of Bacteroidetes isolated from human gut microbiota
The authors analyzed and annotated the data from the CAZy database by determining the whole genome sequence of the gut microbiota. Four unreported Bacteroides were discovered in the heparinase genes. Further studies showed that all four Bacteroides could efficiently utilize heparin in vitro, and all four new heparinases expressed had good enzymatic activity.
References and literature reviewed were also fine.
Figures and tables were adequate, with good discussion and necessarily concluded.
Exceptionally well represented and structured, with references
Author Response
Thanks for your suggestions, please refer to the attachment for details of the changes.

Reviewer 2 Report
The manuscript „A novel thermostable glycosaminoglycan lyase based on the genome of Bacteroidetes isolated from human gut microbiota” reports detailed characterization of enzymes that hydrolyze heparin (a relatively abundant human glycosaminoglycan). The study shows that 4 Bacteroides spp. selected based on their genomic sequence (I presume) can grow on heparin as sole carbon source. By using synthetic gene sequences the enzymes were produced in a recombinant and caracterised in terms of thermal stability, pH and temperature optimum. Finally 3 D structure of enzymes were computed.
The manuscript is overall well written, and is based on a large amount of experimental data. However, it the manuscript deals with a subject that does not seem to be within the scope of the journal and is rather difficult to follow for a reader interested in foods. The data presented in the manuscript appears to be more suitable for a biotechnology journal.
In the section 3.3. bacterial names are occasionslly fully written and occasionsally abbreviated with two letter abbreviations (which are intuitive but never introduced). This should be synchronized (use either of the two forms)
The section 3.1. Screening of novel heparin-degrading strains should be written in Materials and Methods section, and it is suggested to present Table 1 as supplementary material.
In the results section 3.2. please clarify what you mean by “we annotated the bacteriophage genome with CAZymes” The methods section does not report any analysis related to bacteriophages!
In the sections 2.7 and 3.5. the host for the vector insertion should be specified. It is stated that synthetic gene sequence was cloned, while the Table 2 gives primer sequences. This procedure should be explained with sufficient detail.
Please justify defining kinetic parameters at 37C and pH 7.4, given that different enzymes had different optimal temperatures and pH.
Please integrate in one paragraph the procedure use to obtain purified enzymes suitable for determining 3D structure
Table legend is not sufficiently detailed and informative (for tables 2-4), while relevant info is missing (eg. in Table 2 it is unclear how these primer sequences were selected)
Minor comments
Avoid using flora instead of microbiota (in the abstract Fig 1 legend and elsewhere)
Define “4,5-no oligosaccharides” (mentioned in the introduction)
Parentheses in Materials and Methods “(screened and saved” are not closed
The use of phrase “anaphylactic bacteria” is not appropriate, revise
Author Response

(The authors gave the same response as above.)

Reviewer 3 Report
In this manuscript, Chuan Zhang and cols. explore the capacity of Bacteroides strains to use the glycosaminoglycan heparin as a carbon source. They first carried out a comparative analysis of genomic sequences using the CAZy database and the Jiangnan University database to search for the presence of family genes Polysaccharide Lyase Family 13 (PL13) on Bacteroides strains. They can identify four Bacteroides strains containing PL13 genes and able to use glycosaminoglycan as carbon source. The authors then investigate the properties of these four enzymes more in detail. Based on their results he authors claim that one of the four enzymes has strong enzymatic activity and significant thermal stability. The work is very interesting and pleasant to read, however, several points should be clearer to avoid confusion.
Main points
The methodology explained in the abstract section seems different to explained in the text, this create confusion (marked in green in the attached document). For example:
Is not very clear if the four mentioned Bacteroides species were obtained as the results of database screening of genomes or if the strains were first isolated, sequenced and then used to analyses for the presence of PL13 genes.
Is not totally clear if they screened 4, 11, 10 or 17 strains ?
Is not totally clear if the four mentioned Bacteroides strains are unreported or previously reported ?
I suggest that an organization of the described elements can allow a better understanding of the work.
I suggest adapting the title to the results of the work for a better understanding.
Is necessary repeat glycosaminoglycan (heparin) each time?
it is a context of “human temperature” it is difficult to understand why the optimal temperature to enzyme function is high as 40-50°, could you discuss more about it?
Did you use a positive control for the enzyme activity assays? what were the results? please add information about controls
Minor points are marked in yellow in the attached document

Author Response
Thanks for your suggestions, please refer to the attachment for details of the changes

Round 2
Reviewer 2 Report
The revised manuscript written by Zhang and coauthors represents an extended version of the originally submitted manuscript, with more technical details. I still stand at the position that the manuscript should be transferred to another journal and revised by reviewers more competent critically address some of the presented data, as at some points data is technically confusing , as detailed for some of the issues in the submitted comments:
Line 99 – it is stated that the sequencing assembly information is shown in Table 1, but Table 1 reports primers.
Line 130 In the expanded section regarding “Cloning, expression, and purification of heparinase”, finally the host for cloning is reported. However, PCR conditions are not defined. In relation to that in Table 1 primers are exceptionally long (the length of primer pair for “amino-acid change” Bc-HepI is 39nt for FOR and 46 nt for REV primer). These “primers” have annealing temperature over 80C, which is higher than polymerase working temperature, and therefore cannot be applied in PCR!
Line 149-153 – it is very strange that elution buffer has the same composition as the washing buffer.
Line 151 – the Table 1 legend is still incomplete – why some parts of primer sequence are not given in capital letters? Why is it relevant to mark “Restrictive restriction sites”? What is “amino-acid change” in the Table 1 title?
Line 243 In Fig 1 – what is the color code on Fig1 a (cluster heatmap of CAZymes of common intestinal microorganisms)? What are PL, AA, CBP etc. abbreviations standing for?
Overall the manuscript presents a lot of data which are challenging to follow for a reader of “Foods”, and which at some instances are not reported with sufficient detail to appear technically sound.
